# Meta-Auto-Decoder for Solving Parametric Partial Differential Equations

**Xiang Huang**[*]
sahx@mail.ustc.edu.cn
University of Science and
Technology of China

**Zhanhong Ye**[*]
yezhanhong@pku.edu.cn
Peking University

**Hongsheng Liu**
liuhongsheng4@huawei.com
Huawei Technologies Co. Ltd

**Beiji Shi**
shibeiji@huawei.com
Huawei Technologies Co. Ltd

**Zidong Wang**
wang1@huawei.com
Huawei Technologies Co. Ltd

**Kang Yang**
yangkang22@huawei.com
Huawei Technologies Co. Ltd

**Yang Li**
liyang477@huawei.com
Huawei Technologies Co. Ltd

**Min Wang**
wangmin106@huawei.com
Huawei Technologies Co. Ltd

**Haotian Chu**
chuhaotian2@huawei.com
Huawei Technologies Co. Ltd

**Fan Yu**
fan.yu@huawei.com
Huawei Technologies Co. Ltd

**Bei Hua**
bhua@ustc.edu.cn
University of Science and
Technology of China

**Lei Chen**
leichen@cse.ust.hk
Hong Kong University of
Science and Technology

**Bin Dong**[✉]
dongbin@math.pku.edu.cn
Beijing International Center for Mathematical Research, Peking University
Center for Machine Learning Research, Peking University

## Abstract

Many important problems in science and engineering require solving the so-called parametric partial differential equations (PDEs), i.e., PDEs with different physical parameters, boundary conditions, shapes of computation domains, etc. Recently, building learning-based numerical solvers for parametric PDEs has become an emerging new field. One category of methods such as the Deep Galerkin Method (DGM) and Physics-Informed Neural Networks (PINNs) aim to approximate the solution of the PDEs. They are typically unsupervised and mesh-free, but require going through the time-consuming network training process from scratch for each set of parameters of the PDE. Another category of methods such as Fourier Neural Operator (FNO) and Deep Operator Network (DeepONet) try to approximate the solution mapping directly. Being fast with only one forward inference for each PDE parameter without retraining, they often require a large corpus of paired input-output observations drawn from numerical simulations, and most of them need a predefined mesh as well. In this paper, we propose Meta-Auto-Decoder (MAD), a mesh-free and unsupervised deep learning method that enables

---

[*]The first two authors contributed equally to this paper, and Bin Dong is the corresponding author. Zhanhong Ye proposed MAD-L and explain the effectiveness of the MAD method from the perspective of manifold learning. Huang Xiang proposed MAD-LM on the basis of MAD-L and completed all the experiments in the paper. Xiang Huang performed this work during an internship at Huawei.

the pre-trained model to be quickly adapted to equation instances by implicitly encoding (possibly heterogenous) PDE parameters as latent vectors. The proposed method MAD can be interpreted by manifold learning in infinite-dimensional spaces, granting it a geometric insight. Extensive numerical experiments show that the MAD method exhibits faster convergence speed without losing accuracy than other deep learning-based methods. The project page with code is available: https://gitee.com/mindspore/mindscience/tree/master/MindElec/.

# 1 Introduction

Many important problems in science and engineering, such as inverse problems, control and optimization, risk assessment, and uncertainty quantification [1, 2], require solving the so-called parametric PDEs, i.e., partial differential equations (PDEs) with different physical parameters, boundary conditions, or solution regions. Mathematically, they require to solve the so-called *parametric* PDEs that can be formulated as:

$$\mathcal{L}_{\widetilde{x}}^{\gamma_1} u = 0, \ \widetilde{x} \in \Omega \subset \mathbb{R}^d, \qquad \mathcal{B}_{\widetilde{x}}^{\gamma_2} u = 0, \ \widetilde{x} \in \partial\Omega \tag{1}$$

where $\mathcal{L}^{\gamma_1}$ and $\mathcal{B}^{\gamma_2}$ are partial differential operators parametrized by $\gamma_1$ and $\gamma_2$, respectively, and $\widetilde{x}$ denotes the independent variable in spatiotemporal-dependent PDEs. Given $\mathcal{U} = \mathcal{U}(\Omega; \mathbb{R}^{d_u})$ and the space of parameters $\mathcal{A}$, $\eta = (\gamma_1, \gamma_2, \Omega) \in \mathcal{A}$ is the variable parameter of the PDEs and $u \in \mathcal{U}$ is the solution of the PDEs. Note that the form of $\eta$ considered here is very general with possible heterogeneity allowed, since the computational domain shape $\Omega$ and the functions defined on this domain or its boundary (which may be involved in $\gamma_1, \gamma_2$) is obviously of different type. Solving parametric PDEs requires to learn an infinite-dimensional operator $G : \mathcal{A} \to \mathcal{U}$ that map any PDE parameter $\eta$ to its corresponding solution $u^\eta$ (i.e., the solution mapping).

In recent years, learning-based PDE solvers have become very popular, and it is generally believed that learning-based PDE solvers have the potential to improve efficiency [3, 4, 5]. The learning-based PDE solvers can be categorized into two categories in terms of the objects that are approximated by neural networks (NN), i.e., the approximation of the solution $u^\eta$ and the approximation of the solution mapping $G$.

**NN as a new ansatz of solution.** This kind of approaches approximate the solution of the PDEs with a neural network and mainly rely on governing equations and boundary conditions (or their variants) to train the neural networks. For example, PINNs [3] and DGM [6] constrain the output of deep neural networks to satisfy the given governing equations and boundary conditions. Deep Ritz Method (DRM) [7] exploits the variational form of PDEs and can be used to solve PDEs that can be reformulated as equivalent energy minimization problems. Based on a weak formulation of PDEs, Weak Adversarial Network (WAN) [8] parameterizes the weak solution and test functions as primal and adversarial neural networks, respectively. These neural approximation methods can work in an unsupervised manner, without the need to generate labeled data from conventional computational methods. However, all these methods treat different PDE parameters as independent tasks, and need to retrain the neural network from scratch for each PDE parameter. When a large number of tasks with different PDE parameters need to be solved, these methods are computationally expensive and impractical. In order to mitigate retraining cost, E and Yu [7] recommends a transfer learning method that uses a model trained for one task as the initial model to train another task. However, according to our experiments, transfer learning method is not always effective in improving convergence speed (see Sec.3.1, 3.3).

**NN as a new ansatz of solution mapping.** This kind of approaches use neural networks to learn the solution mapping between two infinite-dimensional function spaces [9, 10, 11, 12, 13]. For example, PDE-Nets [9, 10] are among the earliest neural operators that are specifically designed convolutional neural networks inspired by finite difference approximations of PDEs. They are able to uncover hidden PDE models from observed dynamical data and perform fast and accurate predictions at the same time. DeepONet [11] uses two subnets to encode the parameters and location variables of the PDEs separately, and merge them together to compute the solution. FNO [13] utilizes fast Fourier transform to build the neural operator architecture and learn the solution mapping between two infinite-dimensional function spaces. A significant advantage of these approaches is that once the

neural network is trained, the prediction time is almost negligible. Although they have demonstrated promising results across a wide range of applications, several issues occur. First, the data acquisition cost is prohibitive in complex physical, biological, or engineering systems, and the generalization ability of these models is poor when there is not enough labeled data [14]. Second, most of these methods [9, 10, 12, 13] require a predefined mesh and utilize the labeled data on the mesh for training and inference. Third, simply applying one forward inference may lead to unsatisfactory generalization, especially on out-of-distribution (OOD) settings (i.e., PDE parameters for training and inference are from different probability distributions). Finally, these operators directly takes the PDE parameter $\eta$ as network input, which would bring inconvenience in network implementation if $\eta$ is heterogeneous. The recently proposed Physics-Informed DeepONet (PI-DeepONet) [15] can learn a mesh-free solution mapping without any labeled data and retraining. However, it needs to collect a large number of training samples in the parameter space $\mathcal{A}$ to obtain an acceptable accuracy (see Sec.3.1), and is still inflexible dealing with heterogeneous PDE parameters.

**Meta-Learning.**   Different from conventional machine learning that learns to do a given task, meta-learning learns to improve the learning algorithm itself based on multiple learning episodes over a distribution of related tasks. As a result, meta-learning can handle new tasks faster and better. In this field, the Model-Agnostic Meta-Learning (MAML) [16] algorithm and its variants [17, 18, 19] have beed widely used. These algorithms try to find an initial model with good generalization ability such that it can be adapted to new tasks with a small number of gradient updates. For example, MAML [16] firstly trains a meta-model with good initialization weight on a variety of learning tasks, which is then fine-tuned on a new task through a few steps of gradient descent to get the target model. The Reptile [18] algorithm eliminates second-order derivatives in MAML algorithm by repeatedly sampling a task, training on it, and moving the initialization towards the trained weight on that task.

Borrowing the idea of meta-learning may inspire new ways to solve parametric PDEs, where different PDE parameters correspond to different tasks. To the best of our knowledge, Meta-MgNet [20] is the first work that view solving parametric PDEs as a meta-learning problem, which is based on hypernet and the multigrid algorithm. Meta-MgNet utilizes the similarity between tasks to generate good smoothing operators adaptively, and thereby accelerates the solution process, but is not directly applicable to PDEs on which the multigrid algorithm is not available. Recently, the Reptile algorithm is also used to accelerate the PDE solving problems in [21]. However, MAML and Reptile are not always effective in improving convergence speed (see Sec.3.2 and 3.3).

**Our contributions.**   We propose Meta-Auto-Decoder (MAD), a mesh-free and unsupervised deep learning method that enables the pre-trained model to be quickly adapted to equation instances by implicitly encoding heterogeneous PDE parameters as latent vectors. Different from Meta-MgNet, MAD makes use of the similarity between tasks from the perspective of manifold learning, and tries to learn a nonlinear approximation of the solution manifold. We construct the ansatz of solution as a neural network in the form $u_\theta(\widetilde{x}, z)$. By taking the spatial (or spatial-temporal) coordinate $\widetilde{x}$ directly as the network input, unsupervised training loss is allowed, and a mesh is no longer required. As the additional input $z$ varies, $u_\theta(\widetilde{x}, z)$ moves on a manifold in an infinite-dimensional function space, which may be an approximation of the true solution manifold for certain $\theta$. The PDE parameter $\eta$ is implicitly encoded into $z$ by applying the auto-decoder architecture motivated by [22], regardless of the possible heterogeneity. When a new task comes, MAD achieves fast transfer by projecting the new task to the manifold and fine-tuning the manifold at the same time. The main contributions of this paper are summarized as follows:

- A mesh-free and unsupervised deep neural network approach is proposed to solve parametric PDEs. Based on meta-learning concept, once the neural network is pre-trained, solving a new task involves only a small number of iterations. In addition, the auto-decoder architecture adopted by MAD can realize auto-encoding of heterogeneous PDE parameters.

- The mathematical intuition behind the MAD method is analyzed from the perspective of manifold learning. In short, a neural network is pre-trained to approximate the solution manifold, and the required solution is searched on the solution manifold or in a neighborhood of the solution manifold.

- Extensive numerical experiments are carried out to demonstrate the effectiveness of our method, which show that MAD can significantly improve the convergence speed and has good extrapolation ability for OOD settings.

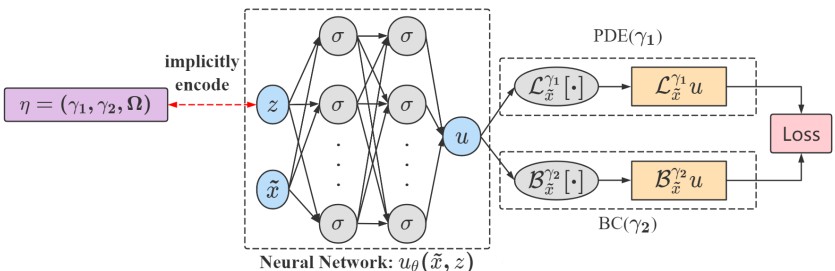

Figure 1: Architecture of Meta-Auto-Decoder.

## 2 Methodology

### 2.1 Meta-Auto-Decoder

We adopt meta-learning concept to realize fast solution of parametric PDEs. Our basic idea is to first learn some universal meta-knowledge from a set of sampled tasks in the pre-training stage, and then solve a new task quickly by combining the task-specific knowledge with the shared meta-knowledge in the fine-tuning stage. We also adapt the auto-decoder architecture in [22], and introduce $u_\theta(\widetilde{x}, z)$ to approximate the solutions of parametric PDEs. The architecture of $u_\theta(\widetilde{x}, z)$ is shown in Fig.1. A physics-informed loss is used for training, making the proposed method unsupervised. Putting all these together, we propose a new method Meta-Auto-Decoder (MAD) to solve parametric PDEs. For the rest of the subsection, the loss function and the two stages of training will be explained in details.

To enable unsupervised learning, given any PDE parameter $\eta \in \mathcal{A}$, the physics-informed loss $L^\eta : \mathcal{U} \to [0, \infty)$ about Eq.(1)

$$L^\eta[u] = \|\mathcal{L}_{\widetilde{x}}^{\gamma_1} u\|_{L^2(\Omega)}^2 + \lambda_{\text{bc}} \|\mathcal{B}_{\widetilde{x}}^{\gamma_2} u\|_{L^2(\partial\Omega)}^2 \tag{2}$$

is considered, where $\lambda_{\text{bc}} > 0$ is a weighting coefficient. The Monte Carlo estimate of $L^\eta[u]$ is

$$\hat{L}^\eta[u] = \frac{1}{M_{\text{r}}} \sum_{j=1}^{M_{\text{r}}} \left\| \mathcal{L}_{\widetilde{x}}^{\gamma_1} u(\widetilde{x}_j^{\text{r}}) \right\|_2^2 + \frac{\lambda_{\text{bc}}}{M_{\text{bc}}} \sum_{j=1}^{M_{\text{bc}}} \left\| \mathcal{B}_{\widetilde{x}}^{\gamma_2} u(\widetilde{x}_j^{\text{bc}}) \right\|_2^2, \tag{3}$$

where $\{\widetilde{x}_j^{\text{r}}\}_{j \in \{1,...,M_{\text{r}}\}}$ and $\{\widetilde{x}_j^{\text{bc}}\}_{j \in \{1,...,M_{\text{bc}}\}}$ are two sets of random sampling points in $\Omega$ and $\partial\Omega$, respectively. This task-specific loss $\hat{L}^\eta[u]$ can be computed by automatic differentiation [23], and will be used in the pre-training stage and the fine-tuing stage.

In the pre-training stage, through minimizing the loss function, a pre-trained model parametrized by $\theta^*$ is learned for all tasks and each task is paired with its own decoded latent vector $z_i^*$. Such a pre-trained model is considered as the meta knowledge as it is learned from the distribution of all tasks and the learned latent vector $z_i^*$ is the task-specific knowledge. When solving a new task in the fine-tuning stage, keep the model weight $\theta^*$ fixed and minimize the loss by fine-tuning the latent vector $z$. Alternatively, we may unfreeze $\theta$ and allow it to be fine-tuned along with $z$. These two fine-tuning strategies give rise to different versions of MAD, which are called *MAD-L* and *MAD-LM*, respectively. The corresponding problems of pre-training and fine-tuning are formulated as follows:

**Pre-training Stage**  Given $N$ randomly generated PDE parameters $\eta_1, \ldots, \eta_N \in \mathcal{A}$, both *MAD-L* and *MAD-LM* solve the following optimization problem

$$(\{z_i^*\}_{i \in \{1,...,N\}}, \ \theta^*) = \underset{\theta, \{z_i\}_{i \in \{1,...,N\}}}{\arg\min} \sum_{i=1}^{N} \left( \hat{L}^{\eta_i}[u_\theta(\cdot, z_i)] + \frac{1}{\sigma^2} \|z_i\|^2 \right), \tag{4}$$

where $\theta^*$ is the optimal model weight, $\{z_i^*\}_{i \in \{1,...,N\}}$ are the optimal latent vectors for different PDE parameters, and $\hat{L}^{\eta_i}$ is defined in Eq.(3). The regularization $\frac{1}{\sigma^2} \|z_i\|^2$ is added for training stability.

**Fine-tuning Stage (*MAD-L*)**   Given a new PDE parameter $\eta_{\text{new}}$, *MAD-L* keeps $\theta^*$ fixed, and minimizes the following loss function to get

$$z_{\text{new}}^* = \arg\min_z \hat{L}^{\eta_{\text{new}}}[u_{\theta^*}(\cdot, z)] + \frac{1}{\sigma^2}\|z\|^2, \tag{5}$$

then $u_{\theta^*}(\cdot, z_{\text{new}}^*)$ is the approximate solution of PDEs with parameter $\eta_{\text{new}}$. To speed up convergence, we can set the initial value of $z$ to $z_i^*$ obtained during pre-training where $\eta_i$ is the nearest[2] to $\eta_{\text{new}}$.

**Fine-tuning Stage (*MAD-LM*)**   *MAD-LM* fine-tunes the model weight $\theta$ with the latent vector $z$ simultaneously, and solves the following optimization problem

$$(z_{\text{new}}^*, \theta_{\text{new}}^*) = \arg\min_{z,\theta} \hat{L}^{\eta_{\text{new}}}[u_\theta(\cdot, z)] + \frac{1}{\sigma^2}\|z\|^2 \tag{6}$$

with initial model weight $\theta^*$. This would produce an alternative approximate solution $u_{\theta_{\text{new}}^*}(\cdot, z_{\text{new}}^*)$. The latent vector is initialized in the same way as *MAD-L*.

**Remark 1** *The MAD method has several key advantages compared with existing methods. Besides being mesh-free and unsupervised, it can deal with heterogeneous PDE parameters painlessly, since $\eta$ is not taken as the network input, and is encoded into $z$ in an implicit way. Introduction of the meta-knowledge $\theta^*$ would accelerate the fine-tuning process, which can be better understood in the light of the manifold learning perspective. For MAD-LM, the accuracy on OOD tasks is likely to be at least comparable with training from scratch based on PINNs. Although the fine-tuning process of MAD is still slower than one forward inference of a neural network solution mapping, the advantages presented above can make it more suitable for some real applications.*

**Remark 2** *If we replace the physics-informed loss by certain supervised loss, the MAD-L method would then coincide with the DeepSDF algorithm [22]. Despite of this, the field of solving parametric PDEs is quite different from 3D shape representation in computer graphics. Moreover, the introduction of model weight fine-tuning in MAD-LM can significantly improve solution accuracy, as is explained intuitively in Sec.2.2,2.3 and validated by numerical experiments in Sec.3.*

## 2.2   Manifold Learning Interpretation of *MAD-L*

We interpret how the *MAD-L* method works from the manifold learning perspective, which also provides a new interpretation of the DeepSDF algorithm [22]. For the rest of this section, the domain $\Omega$ is fixed and excluded from $\eta$ for simplicity. Now, we consider the following scenario.

**Scenario 1** *The set of solutions $G(\mathcal{A}) = \{G(\eta) \mid \eta \in \mathcal{A}\} \subset \mathcal{U}$ is contained in a low-dimensional structure. To be more specific, there is a finite-dimensional space $Z = \mathbb{R}^l$ (with $l \ll \dim\mathcal{U}$) and a Lipschitz continuous mapping $\bar{G} : Z \to \mathcal{U}$, such that $G(\mathcal{A}) \subseteq \bar{G}(Z)$. In other words, for any $\eta \in \mathcal{A}$, there exists $z \in Z$ satisfying $\bar{G}(z) = G(\eta)$.*

The mapping $\bar{G}$ is Lipschitz continuous if and only if there exists some $C > 0$ such that $\|\bar{G}(z) - \bar{G}(z')\|_{\mathcal{U}} \le C\|z - z'\|$ for all $z, z' \in Z$. This Lipschitz continuous constraint excludes highly irregular mappings like space-filling curves. When $\mathcal{A}$ is a finite-dimensional space and $G$ is Lipschitz continuous, the parametric PDE would fall into this scenario (just take $Z = \mathcal{A}$, $\bar{G} = G$). Since $\dim Z \ll \dim\mathcal{U}$ (the latter is usually infinity) holds, we may view the mapping $\bar{G}$ as some sort of "decoder", and $Z$ is the corresponding latent vector space, despite of the fact that there doesn't exist an "encoder". In many cases, $\bar{G}(Z) \subset \mathcal{U}$ forms an embedded submanifold, and therefore our MAD method can be viewed as a manifold-learning approach. Once the mapping $\bar{G}$ is learned as above, then for a given parameter $\eta$, searching for the solution $u^\eta$ in the whole space $\mathcal{U}$ is no longer needed. Instead, we may focus on the smaller subset $\bar{G}(Z)$, i.e. the class of functions in $\mathcal{U}$ that is parametrized by $Z$, since $u^\eta = G(\eta) \in \bar{G}(Z)$ holds for any $\eta \in \mathcal{A}$. We then solve the optimization problem

$$z^\eta = \arg\min_z L^\eta[\bar{G}(z)], \tag{7}$$

---

[2]For example, if $\mathcal{A}$ is a space of functions, we can discretize a function into a vector and then find the Euclidean distance between the two vectors as the distance between two PDE parameters.

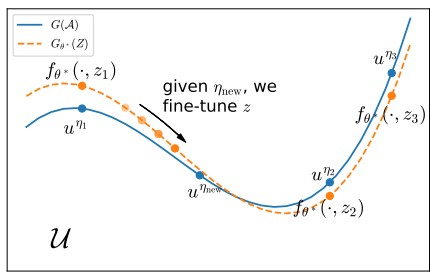

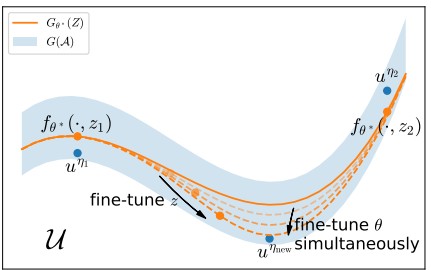

(a) Fine-tune $z$ only (*MAD-L*)    (b) Fine-tune both $z$ and $\theta$ (*MAD-LM*)

Figure 2: Illustration of how MAD works from the manifold learning perspective. **(a) *MAD-L*:** The function space $\mathcal{U}$ is mapped to a 2-dimensional plane. The blue solid curve represents the solution set $G(\mathcal{A})$ formed by exact solutions corresponding to all possible PDE parameters, and each point on the curve represents an exact solution corresponding to one PDE parameter. The orange dotted curve represents the solution set $G_{\theta^*}(Z)$ obtained by the pre-trained model, and each point on the curve corresponds to a latent vector $z$. Given $\eta_{\text{new}} \in \mathcal{A}$, rather than searching in the entire function space $\mathcal{U}$, *MAD-L* only searches on the orange dotted curve to find an optimal $z$ such that its corresponding solution $u_{\theta^*}(\cdot, z)$ is nearest to the blue point $u^{\eta_{\text{new}}}$. **(b) *MAD-LM*:** The solution set $G(\mathcal{A})$ lies within a neighborhood of $G_{\theta^*}(Z)$ that is represented by a gray shadow band. To find solution $u^{\eta_{\text{new}}}$, we have to fine-tune $\theta$ (i.e., the orange dotted lines) and the latent vector $z$ (i.e., the points on the orange dotted lines) simultaneously to approach the exact solution. As the search scope is limited to a strip with a small width, the fine-tuning process can be expected to converge quickly.

and $\bar{G}(z^\eta)$ is the approximate solution. Assuming that the dimension of $Z$ is chosen (either empirically or through trial and error), the aim is to find the mapping $\bar{G}$. Since such a mapping is usually complex and hard to design by hand, we consider the $\theta$-parametrized[3] version $G_\theta : Z \to \mathcal{U}$, and find the best $\theta$ automatically by solving an optimization problem. $G_\theta$ can be constructed in the simple form

$$G_\theta(z)(\widetilde{x}) = u_\theta(\widetilde{x}, z), \tag{8}$$

where $u_\theta$ is a neural network whose input is the concatenation of $\widetilde{x} \in \mathbb{R}^d$ and $z \in \mathbb{R}^l$. The next step is to find the optimal model weight $\theta$ via training, with the target being $G(\mathcal{A}) \subseteq G_\theta(Z)$. Assuming that the PDE parameters are generated from a probability distribution $\eta \sim p_\mathcal{A}$, then $G(\eta) \in G_\theta(Z)$ holds almost surely if and only if

$$d(\theta) = \mathop{\mathbb{E}}_{\eta \sim p_\mathcal{A}} \left[ d_\mathcal{U}\big(u^\eta, G_\theta(Z)\big) \right] = \mathop{\mathbb{E}}_{\eta \sim p_\mathcal{A}} \left[ \min_z \big\| u^\eta - u_\theta(\cdot, z) \big\|_\mathcal{U} \right] = 0, \tag{9}$$

which suggests taking $\theta^* = \arg\min_\theta d(\theta)$. In case we do not have direct access to the exact solutions $u^\eta$, the equivalent[4] condition

$$d'(\theta) = \mathop{\mathbb{E}}_{\eta \sim p_\mathcal{A}} \left[ \min_z L^\eta[u_\theta(\cdot, z)] \right] = 0 \tag{10}$$

is considered, and $d'(\theta)$ becomes the alternative loss to be minimized. In the specific implementation, the expectation on $\eta \sim p_\mathcal{A}$ is estimated by Monte Carlo samples $\eta_1, \ldots, \eta_N$, and the optimal network weight $\theta$ is taken to be

$$\theta^* \approx \arg\min_\theta \frac{1}{N} \sum_{i=1}^N \min_{z_i} L^{\eta_i}[u_\theta(\cdot, z_i)]. \tag{11}$$

We further estimate the physics-informed loss $L^\eta$ using Monte Carlo method to obtain Eq.(4). After that, when a new PDE parameter $\eta_{\text{new}} \in \mathcal{A}$ comes, a direct adaptation of Eq.(7) would then give rise to the fine-tuning process Eq.(5), since $u_{\theta^*}(\cdot, z) = G_{\theta^*}(z) \approx G(z)$ holds. An intuitive illustration of how *MAD-L* works from the manifold learning perspective is given in Fig.2(a).

---

[3]Two types of parametrization are considered here. The latent vector $z$ parametrizes a point on the manifold $\bar{G}(Z)$ or $G_\theta(Z)$, and $\theta$ parametrizes the shape of the entire manifold $G_\theta(Z)$.

[4]Assume that the solution of Eq.(1) is unique for all $\eta \in \mathcal{A}$, and $u \in \mathcal{U}$ is the solution if and only if $L^\eta[u] = 0$.

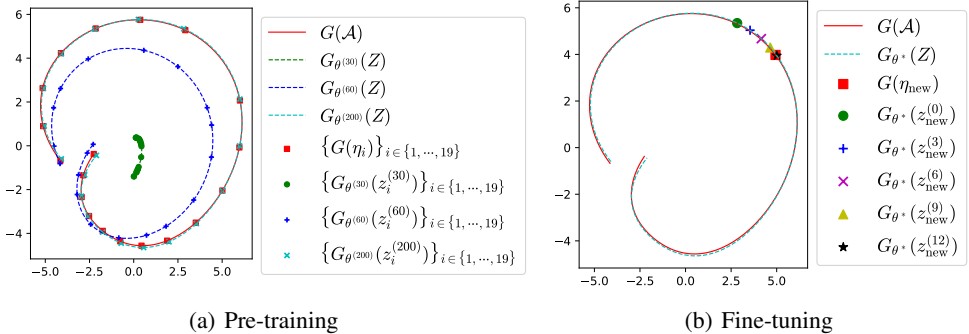

(a) Pre-training                      (b) Fine-tuning

Figure 3: Visualization of the MAD pre-training and fine-tuning process for the ODE problem.

**A Visualization Example** An ordinary differential equation (ODE) is used to visualize the pre-training and fine-tuning processes of *MAD-L*. Consider the following problem with domain $\Omega = (-\pi, \pi) \subset \mathbb{R}$:

$$\frac{\mathrm{d}u}{\mathrm{d}x} = 2(x - \eta)\cos\big((x - \eta)^2\big), \qquad u(\pm\pi) = \sin\big((\pm\pi - \eta)^2\big). \tag{12}$$

We sample 20 points equidistantly on the interval $[0, 2]$ as variable ODE parameters, and randomly select one $\eta_{\text{new}}$ for fine-tuning stage and the rest $\{\eta_i\}_{i \in \{1, \cdots, 19\}}$ for pre-training stage. *MAD-L* generates a sequence of $(\theta^{(m)}, \{z_i^{(m)}\}_{i \in \{1, \cdots, 19\}})$ in pre-training stage, and terminates at $m = 200$ with the optimal $(\theta^*, \{z_i^*\}_{i \in \{1, \cdots, 19\}})$. The infinite-dimensional function space $\mathcal{U} = C([-\pi, \pi])$ is projected onto a 2-dimensional plane using Principal Component Analysis (PCA). Fig.3(a) visualizes how $G_\theta(Z)$ gradually fits $G(\mathcal{A})$ in pre-training stage. The set of exact solutions $G(\mathcal{A})$ forms a 1-dimensional manifold (i.e. the red solid curve), and the marked points $\{G(\eta_i)\}_{i \in \{1, \cdots, 19\}}$ represent the corresponding ODE parameters used for pre-training. Each dotted curve represents a solution set $G_\theta^{(m)}(Z)$ obtained by the neural network at the $m$-th iteration with the points $G_{\theta^{(m)}}(z_i^{(m)}) = u_{\theta^{(m)}}(\cdot, z_i^{(m)})$ also marked on the curve. As the number of iterations $m$ increases, the network weight $\theta = \theta^{(m)}$ updates, making the dotted curves evolve and finally fit the red solid curve, i.e., the target manifold $G(\mathcal{A})$. Fig.3(b) illustrates the fine-tuning process for a given new ODE parameter $\eta_{\text{new}} \in \mathcal{A}$. As in Fig.3(a), the red solid curve represents the set of exact solutions $G(\mathcal{A})$, while the cyan dotted curve represents the solution set $G_{\theta^*}(Z) = G_{\theta^{(200)}}(Z)$ obtained by the pre-trained network. As $z = z_{\text{new}}^{(m)}$ updates (i.e., through fine-tuning $z$), the marked point $G_{\theta^*}(z_{\text{new}}^{(m)})$ moves on the cyan dotted curve, and finally converges to the approximate solution $G_{\theta^*}(z_{\text{new}}^*) = G_{\theta^*}(z_{\text{new}}^{(12)}) \approx G(\eta_{\text{new}})$.

### 2.3 Manifold Learning Interpretation of *MAD-LM*

The *MAD-L* method is designed for Scenario 1. However, many parametric PDEs encountered in real applications do not fall into this scenario, especially when the parameter set $\mathcal{A}$ of PDEs is an infinite-dimensional function space. Simply applying *MAD-L* method to these PDE solving problems would likely lead to unsatisfactory results. However, *MAD-LM* works in a more general scenario, and thus has the potential of getting improved performance for a wide range of parametric PDE problems. This alternative scenario is given as follows.

**Scenario 2** *The solution set $G(\mathcal{A}) \subset \mathcal{U}$ can be approximated by a set with low-dimensional structure, in the sense that there is a finite-dimensional space $Z = \mathbb{R}^l$ (with $l \ll \dim \mathcal{U}$) and a Lipschitz continuous mapping $\bar{G} : Z \to \mathcal{U}$, such that $G(\mathcal{A})$ is contained in the c-neighborhood of $\bar{G}(Z) \subset \mathcal{U}$, where c is a relatively small constant. In other words, for any $\eta \in \mathcal{A}$, there exists some $z \in Z$ satisfying $\|\bar{G}(z) - G(\eta)\|_\mathcal{U} \le c$.*

Appendix A gives an example of $G(\mathcal{A})$ that falls into Scenario 2 but not Scenario 1. In this new scenario, similar derivation leads to the same pre-training stage, which is used to find the initial decoder mapping $G_{\theta^*} \approx \bar{G}$. However, in the fine-tuning stage, simply fine-tuning the latent vector $z$ won't give a satisfactory solution in general due to the existence of the c-gap. Therefore, we have

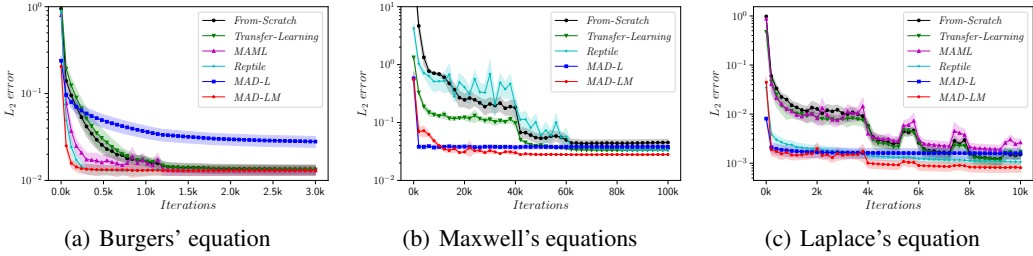

|                          |                           |                          |
| :----------------------: | :-----------------------: | :----------------------: |
| (a) Burgers' equation    | (b) Maxwell's equations   | (c) Laplace's equation   |

Figure 4: The mean $L_2\ error$ convergence with respect to the number of training iterations.

to fine-tune the model weight $\theta$ with the latent vector $z$ simultaneously, and solve the optimization problem Eq.(6). It produces a new decoder $G_{\theta^*_{\text{new}}}(z^*_{\text{new}})$ specific to the parameter $\eta_{\text{new}}$. An intuitive illustration is given in Fig.2(b).

## 3 Numerical Experiments

To evaluate the effectiveness of the MAD method, we apply it to solve three parametric PDEs: (1) Burgers' equation with variable initial conditions; (2) Maxwell's equations with variable equation coefficients; and (3) Laplace's equation with variable solution domains and boundary conditions (heterogeneous PDE parameters). Accuracy of the model is measured by $average\ relative\ L_2\ error$ (abbreviated as $L_2\ error$) between predicted solutions and reference solutions, and we provide the mean value and the 95% confidence interval of $L_2\ error$. We compare MAD with other methods including learning from scratch (abbreviated as *From-Scratch*), *Transfer-Learning* [7], *MAML* [16, 17], *Reptile* [18] and *PI-DeepONet* [15]. For each experiment, the PDE parameters are divided into two sets: $S_1$ and $S_2$. Parameters in $S_1$ correspond to sample tasks for pre-training, and parameters in $S_2$ correspond to new tasks for fine-tuning. See Appendix B for the default experimental setup, and more detailed experimental setups and results for Burgers' equation, Maxwell's equation and Laplace's equation are given in Appendix C, D and E respectively. Unless otherwise specified, all the experiments are conducted under the MindSpore[5].

### 3.1 Burgers' Equation

We consider the 1-D Burgers' equation:

$$\frac{\partial u}{\partial t} + u\frac{\partial u}{\partial x} = \nu\frac{\partial^2 u}{\partial x^2},\ x \in (0,1), t \in (0,1], \qquad u(x,0) = u_0(x),\ x \in (0,1), \qquad (13)$$

Eq.(13) can model one-dimensional flow of a viscous fluid, where $u$ is the velocity, $\nu$ is the viscosity coefficient and initial condition $u_0(x)$ is the changing parameter of the PDE, i.e. $\eta = u_0(x)$. The initial condition $u_0(x)$ is generated using Gaussian random field (GRF) [24] according to $u_0(x) \sim \mathcal{N}(0; 100(-\Delta + 9I)^{-3})$ with periodic boundary conditions.

Fig.4(a) shows the mean $L_2\ error$ of all methods as the number of training iterations increases. All methods converge to nearly the same accuracy (the mean $L_2\ error$ close to 0.013) except for *MAD-L*, which we guess probably due to the $c$-gap introduced in Sec.2.3. In terms of convergence speed, *From-Scratch* and *Transfer-Learning* need about 1200 iterations to converge, whereas *MAML*, *Reptile* and *MAD-LM* need about 200 iterations to converge. *MAD-LM* has the fastest convergence speed, requiring only 17% of the training iterations of *From-Scratch*. In this experiment, *Transfer-Learning* does not show any advantage over *From-Scratch*, which means that *Transfer-Learning* fails to obtain any useful knowledge in pre-training stage.

*PI-DeepONet* can directly inference for unseen PDE parameters in $S_2$, so it has no fine-tuning process. Table 1 shows the comparison of the mean $L_2\ error$ of *PI-DeepONet* and MAD under different numbers of training samples in $S_1$. The results show that *PI-DeepONet* has a strong dependence on the number of training samples, and its mean $L_2\ error$ is remarkably high when $S_1$ is small. Moreover, its mean $L_2\ error$ is significantly higher than that of *MAD-L* or *MAD-LM* in all cases.

---

[5]https://www.mindspore.cn/

Table 1: The mean $L_2\ error$ of *PI-DeepONet* and MAD under different numbers of samples in $S_1$.

| $\|S_1\|$ | PI-DeepONet | MAD-L | MAD-LM |
|---|---|---|---|
| 10 | 0.715 | 0.365 | 0.015 |
| 50 | 0.247 | 0.046 | 0.013 |
| 100 | 0.217 | 0.028 | 0.013 |
| 200 | 0.169 | 0.020 | 0.013 |
| 300 | 0.181 | 0.018 | 0.013 |
| 400 | 0.183 | 0.019 | 0.013 |

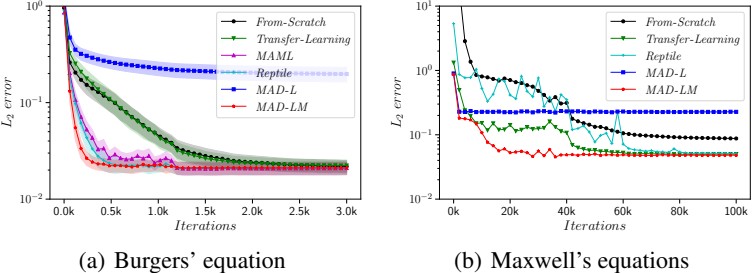

(a) Burgers' equation      (b) Maxwell's equations

Figure 5: The mean $L_2\ error$ convergence with respect to the number of training iterations for extrapolation experiments.

In the above experiments, $\eta$s in $S_1$ and $S_2$ come from the same GRF, so we can assume that the tasks in the pre-training stage come from the same task distribution as the tasks in the fine-tuing stage. We investigate the extrapolation capability of MAD, that is, tasks in the fine-tuing stage come from different task distribution than those in the pre-training stage. Specifically, $S_1$ is still the same as above, but $S_2$ is generated from $\mathcal{N}(0; 100(-\Delta + 25I)^{-2.5})$. Fig.5(a) shows the results of extrapolation experiments. Since the distribution of tasks has changed, the manifold learned in the pre-training stage fits $G(\mathcal{A})$ worse, so *MAD-L* exhibits worse accuracy than that in Fig.4(a). However, as with Fig.4(a), the convergence speed of *MAD-LM* in Fig.5(a) is also better than other methods. This shows that the extrapolation capability of MAD is also better than other methods in this example.

## 3.2 Time-Domain Maxwell's Equations

We consider the time-domain 2-D Maxwell's equations with a point source in the transverse Electric (TE) mode [25]:

$$\frac{\partial E_x}{\partial t} = \frac{1}{\epsilon_0 \epsilon_r}\frac{\partial H_z}{\partial y}, \quad \frac{\partial E_y}{\partial t} = -\frac{1}{\epsilon_0 \epsilon_r}\frac{\partial H_z}{\partial x}, \quad \frac{\partial H_z}{\partial t} = -\frac{1}{\mu_0 \mu_r}\left(\frac{\partial E_y}{\partial x} - \frac{\partial E_x}{\partial y} + J\right), \quad (14)$$

where $E_x$, $E_y$ and $H_z$ are the electromagnetic fields, $J$ is the point source term. The equation coefficients $\epsilon_0$ and $\mu_0$ are the permittivity and permeability in vacuum, respectively. The equation coefficients $\epsilon_r$ and $\mu_r$ are the relative permittivity and relative permeability of the media, respectively. [26] uses modified PINNs method to solve Eq.(14) with fixed $\epsilon_r = 1$ and $\mu_r = 1$. However, in this paper, $(\epsilon_r, \mu_r)$ are variable parameters of the PDEs, i.e., $\eta = (\epsilon_r, \mu_r)$, which corresponds to the media properties in the simulation region.

Fig.4(b) shows that all methods converge to similar accuracy (mean $L_2\ error$ is close to 0.04), and *MAD-LM* achieves the lowest mean $L_2\ error$ (0.028). In terms of convergence speed, *MAD-L* and *MAD-LM* are obviously superior to other methods. It is worth noting that *MAML* fails to converge in pre-training stage, therefore its data is missing in Fig.4(b). We guess the reason is that singularity brought by point source and computation of second-order derivatives pose great difficulties in solving optimization problem. *Reptile* also does not show good generalization ability probably due to the same singularity problem.

We do an extrapolation experiment, where $(\epsilon_r, \mu_r)$ in $S_1$ comes from $[1, 5]^2$, but in the fine-tuning stage, we only consider the $(\epsilon_r, \mu_r) = (7, 7)$ case. Because the extrapolated task does not lie in the task distribution in the pre-training stage, the point $G(\eta_{\text{new}})$ corresponding to the extrapolated task in

the function space $\mathcal{U}$ is not on the learned manifold $G_{\theta^*}(Z)$, which causes *MAD-L* to converge to poor accuracy. Fig.5(b) indicates that *MAD-LM* is significantly faster than *From-Scratch* and *Reptile* in convergence speed while maintaining high accuracy. Notably, *Transfer-Learning* also exhibits faster convergence than *From-Scratch* and *Reptile*. This is because $(\epsilon_r, \mu_r) = (4, 5)$ is randomly selected in the pre-training stage and is very close to $(\epsilon_r, \mu_r) = (7, 7)$ in Euclidean distance.

### 3.3 Laplace's Equation

We consider the 2-D Laplace's equation as follows:

$$\frac{\partial^2 u}{\partial x^2} + \frac{\partial^2 u}{\partial y^2} = 0, \ (x, y) \in \Omega, \qquad u(x, y) = g(x, y), \ (x, y) \in \partial\Omega, \tag{15}$$

where the shape of $\Omega$ and boundary condition $g(x, y)$ are the variable parameters of the PDE, i.e. $\eta = (\Omega, g(x, y))$. In this experiment, we use triangular domain $\Omega$ and vary the shape of $\Omega$ by randomly choosing three points on the circumference of a unit circle to form the triangle. Given that $h$ is the boundary condition on the unit circle, we use GRF to generate $h \sim \mathcal{N}(0, 10^{3/2}(-\Delta + 100I)^{-3})$ with periodic boundary conditions. The analytical solution of the Laplace's equation on the unit circle can be obtained by Fourier method. Then, we use the analytical solution on the three sides of the triangle as the boundary condition $g(x, y)$. The variable PDE parameters include the shape of the solution domain (the shape of the triangle) and the boundary conditions on the three sides of the triangle, so the PDE parameters here are heterogeneous. MAD can implicitly encode such heterogeneous PDE parameters as latent vectors conveniently, whereas *PI-DeepONet* is unable to handle this case without further adaptations.

Fig.4(c) shows that all methods finally converge to similar accuracy (mean $L_2\ error$ close to 0.001), and *MAD-LM* achieves the lowest mean $L_2\ error$. *MAD-L*, *MAD-LM* and *Reptile* show good generalization capability and excellent convergence speed, whereas *Transfer-Learning* and *MAML* do not show any advantage over *From-Scratch*.

**Summary of experimental results.** Achieving fast adaptation is the major focus of this paper, and solutions within a reasonable precision need to be found. Indeed, in many control and inverse problems, a higher precision in solving the forward problem (such as parametric PDEs) does not always lead to better results. For example, a solution with about $5\%$ relative error is already enough for Maxwell's equations in certain engineering scenarios. We are therefore interested in reducing the cost of solving the PDE with a new set of parameters by using only a relatively small number of iterations, in order to obtain an accurate enough solution in practice. The advantages of MAD (especially *MAD-LM*) is directly validated in the numerical experiments, as it achieves very fast convergence in the early stage of the training process. Some other applications may focus more on the final precision, and is not as sensitive to the training cost. In this alternate criterion, the superiority of the MAD method becomes less obvious in our test cases except Maxwell's equations, but its performance is still comparable to other methods.

## 4 Conclusions

In this paper, a novel mesh-free and unsupervised deep learning method MAD is proposed for solving parametric PDEs based on meta-learning idea. A good initial model is obtained in pre-training stage to learn useful information from a set of sampled tasks, which is then used to help solve the parametric PDEs quickly in fine-tuning stage. Moreover, MAD can implicitly encode heterogeneous PDE parameters as latent vectors. The effectiveness of MAD method is analyzed from the perspective of manifold learning and verified by extensive numerical experiments.

## Acknowledgments

This work was supported by National Key R&D Program of China under Grant No. 2021ZD0110400.

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
