# Appendixes

## A  An Example for Scenario 2

We give an example of $G(\mathcal{A})$ which falls into Scenario 2 but not Scenario 1. Let $\{\phi_k(x)\}_{k\in\mathbb{N}}$ be an orthonormal basis of a Hilbert space $L^2(X)$, and $(\lambda_k)_{k\in\mathbb{N}}$ be a sequence of positive real numbers with $\sum_{k=1}^{\infty} \lambda_k < \infty$. We take

$$\mathcal{A} = \left\{ \sum_{k=1}^{\infty} \xi_k \sqrt{\lambda_k} \phi_k(x) \,\middle|\, \xi_k \in [-1,1] \right\} \subset L^2(X), \tag{1}$$

and consider the equation $u_t = -u$ on $\Omega = X \times [0,1]$ with initial condition $u(x,0) = \eta(x)$. It is easy to see that

$$G(\mathcal{A}) = \left\{ \sum_{k=1}^{\infty} \xi_k \sqrt{\lambda_k} \phi_k(x) e^{-t} \,\middle|\, \xi_k \in [-1,1] \right\} \subset \mathcal{U} = L^2(\Omega). \tag{2}$$

and no $(Z, \bar{G})$-pair can make Scenario 1 valid as $G(\mathcal{A})$ is not finite-dimensional. As for Scenario 2, once the number $c$ is given, there exists a large enough $l$ satisfying $\sum_{k=l+1}^{\infty} \lambda_k \leq c^2$. Then we let $Z = \mathbb{R}^l$ and choose a linear mapping $\bar{G}$ such that

$$\bar{G}\big((\xi_k)_{k=1}^{l}\big) = \sum_{k=1}^{l} \xi_k \sqrt{\lambda_k} \phi_k(x) e^{-t}. \tag{3}$$

For any $\eta = \sum_{k=1}^{\infty} \xi_k \sqrt{\lambda_k} \phi_k(x) \in \mathcal{A}$, taking $z = (\xi_k)_{k=1}^{l} \in Z$ gives

$$\begin{aligned}
\big\| \bar{G}(z) - G(\eta) \big\|_{\mathcal{U}}^2 &= \left\| \sum_{k=l+1}^{\infty} \xi_k \sqrt{\lambda_k} \phi_k(x) e^{-t} \right\|_{L^2(X\times[0,1])}^2 \\
&= \sum_{k=l+1}^{\infty} \xi_k^2 \lambda_k \int_X \phi_k(x)^2 \,\mathrm{d}x \int_0^1 e^{-2t} \,\mathrm{d}t \\
&= \sum_{k=l+1}^{\infty} \xi_k^2 \lambda_k \frac{1 - e^{-2}}{2} \\
&\leq \sum_{k=l+1}^{\infty} \lambda_k \\
&\leq c^2.
\end{aligned}$$

This indicates that Scenario 2 is indeed more general than Scenario 1.

## B  Default Experimental Configurations

Below is a detailed explanation of the comparative methods covered in the paper.

- *From-Scratch*: Train the model from scratch based on the PINNs method for all PDE parameters in $S_2$, case-by-case.
- *Transfer-Learning* [7]: Randomly select a PDE parameter in $S_1$ for pre-training stage based on the PINNs method, and then load the obtained weight in pre-training stage for PDE parameters in $S_2$ during fine-tuning stage.
- *MAML* [16, 17]: Meta-train the model for all PDE parameters in $S_1$ based on MAML algorithm. In the meta-testing stage, we load the pre-trained weight $\theta^*$ and fine-tune the model for each PDE parameter in $S_2$.
- *Reptile* [18]: Similar to *MAML*, except that the model weight is updated using the Reptile algorithm in the meta-training stage.
- *PI-DeepONet* [15]: The model is trained based on the training method proposed in [15] for all PDE parameters in $S_1$, and the inference is performed directly for the parameters in $S_2$.
- *MAD-L*: Pre-train the model for all PDE parameters in $S_1$ based on our proposed method and then load and freeze the pre-trained weight $\theta^*$ for the second stage. In the fine-tuning stage, we choose a $z_i^*$ obtained in the pre-training stage to initialize a latent vector for each PDE parameter in $S_2$, and fine-tune the latent vector. The selection of $z_i^*$ is based on the distance between $\eta_{\text{new}}$ and $\eta_i$.
- *MAD-LM*: Different from *MAD-L* that freezes the pre-trained weight, we fine-tune the model weight $\theta$ and the latent vector $z$ simultaneously in the fine-tuning stage.

Unless otherwise specified, the following default configurations are used for the experiments:

- In each iteration, the batch sizes are selected as $(M_r, M_{bc})$ = (8192, 1024).
- For fairness of comparison, the network architectures of all methods (excluding PI-DeepONet) involved in comparison are the same except for the input layer due to the latent vector. For Burgers' equation and Laplace's equation, the standard fully-connected neural networks with 7 fully-connected layers and 128 neurons per hidden layer are taken as a default network structure. For Maxwell's equations, the MS-SIREN network architecture [26] is used that has 4 subnets, each subnet has 7 fully-connected layers and 64 neurons per hidden layer. It is worth noting that our proposed method has gains in different network architectures, and we choose the default network architecture that can achieve high accuracy for the *From-Scratch* method to conduct our comparative experiments.
- The network architecture of *PI-DeepONet* used for Burgers' equation is such that both branch net and trunk net are 7 fully-connected layers and 128 neurons per hidden layer.
- Sine function [27] is used as the activation function, as it exhibits better performance than other alternatives such as ReLU and Tanh.
- The dimension of the latent vector $z$ is determined by trial and error and is set to 128 for Burgers' equation and Laplace's equation, and 16 for Maxwell's equations.
- The Adam optimizer [28] is used with the initial learning rate set to 1e-3 or 1e-4 (whoever achieves the best performance). When the training process reaches 40%, 60% and 80%, the learning rate is attenuated by half.

## C Detailed Experimental Setup and Extended Results for Burgers' Equation

We set the viscosity to $\nu = 0.01$ and solve the Eq.(13) using open source code implemented by [13] to generate the reference solutions. The spatiotemporal mesh size of the ground truth is $1024 \times 101$. In order to solve the Eq.(13) better by using the PINNs-based method, we use the hard constraint on periodic boundary condition mentioned in [29].

We generated 150 different initializations of $u_0(x)$ using GRF and randomly selected 100 cases as $S_1$. The remaining 50 cases are used as $S_2$ for fine-tuning. For *MAD-L* and *MAD-LM*, the pre-training stages run for 50k iterations while the *Transfer-Learning* pre-trains 3k steps since it only handles one single case.

Fig.6 shows model predictions of *MAD-L* and *MAD-LM* compared with the reference solutions under a randomly selected $u_0(x)$ in $S_2$. The predictions of *MAD-L* are in overall approximate agreement

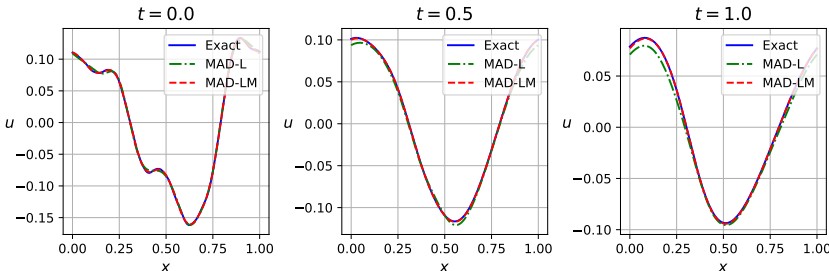

Figure 1: **Burgers' equation:** Reference solutions vs. model predictions at $t = 0.0$, $t = 0.5$ and $t = 1.0$, respectively.

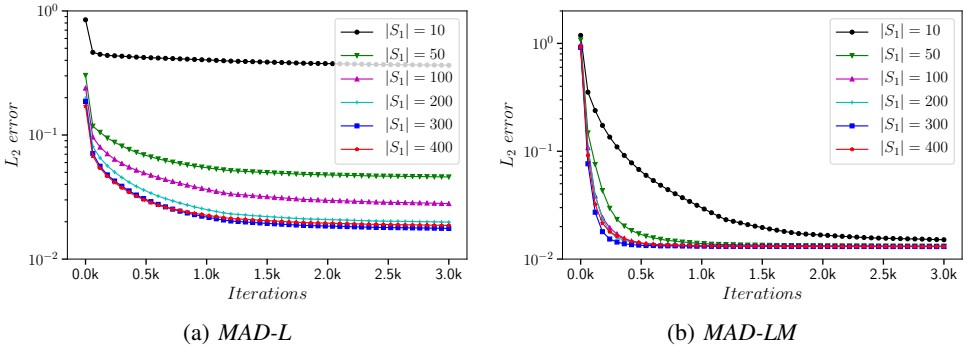

(a) *MAD-L*                (b) *MAD-LM*

Figure 2: **Burgers' equation:** The mean $L_2$ $error$ of *MAD-L* (a) and *MAD-LM* (b) convergence with respect to the number of training iterations under different numbers of samples in $S_1$.

with the reference solutions, but the fit is poor at the spikes and troughs. However, the prediction results of *MAD-LM* is almost the same as that of the reference solutions.

We investigated the effect of the number of samples $|S_1|$ in the pre-training stage on *MAD-L* and *MAD-LM*. Fig.7(a) shows that the accuracy of *MAD-L* after convergence increases with $|S_1|$. However, when $|S_1|$ reaches about 200, increasing $|S_1|$ does not improve the accuracy of the *MAD-L*. This result is consistent with the phenomenon shown in Fig.2(b). More samples in the pre-training stage allow the $G_{\theta^*}(Z)$ to gradually fall within the region formed by $G(\mathcal{A})$. However, when $|S_1|$ reaches a certain level, the $G_{\theta^*}(Z)$ only swings in the region of $G(\mathcal{A})$. Only optimizing $z$ can make the solution move inside the manifold formed by $G_{\theta^*}(Z)$, but $u^{\eta_{\text{new}}}$ may not be close enough to the manifold. Therefore, in order to obtain a more accurate solution, we need to fine-tune $z$ and $\theta$ simultaneously. Fig.7(b) shows that the accuracy and convergence speed of *MAD-LM* do not change significantly with the increase of samples in the pre-training stage. It is only when the number of samples is very small (i.e., $|S_1| = 10$) that the early convergence speed is significantly affected. This shows that *MAD-LM* can perform well in the fine-tuning stage without requiring a large number of samples during the pre-training stage.

For Burgers' equation, we also consider the scenario when the viscosity coefficients $\nu$ in Eq.(13) vary within a certain range, i.e., the PDE parameter $\eta = (\nu, u_0(x))$ is heterogeneous. Specifically, $\nu \in \{10^\beta | \beta \sim U(-3, -1)\}$ where $U$ is a uniform distribution and $u_0(x) \sim \mathcal{N}(0; 100(-\Delta + 9I)^{-3})$. In this experiment, $|S_1| = 100$ and $|S_2| = 50$ while $S_1$ and $S_2$ come from the same task distribution. Fig.8 compares the convergence curves of mean $L_2$ $error$ corresponding to different methods. *MAD-LM* has obvious speed and accuracy improvement over *From-Scratch* and *Transfer-Learning*. It's worth noting that *MAML* and *Reptile* also perform well in this scenario.

We investigated the effect of the dimension of the latent vector (latent size) in Burgers' equation on performance. As can be seen from Fig.9(a), for MAD-L, different latent sizes have different performances and the best performance is achieved when it is equal to 128. As can be seen from

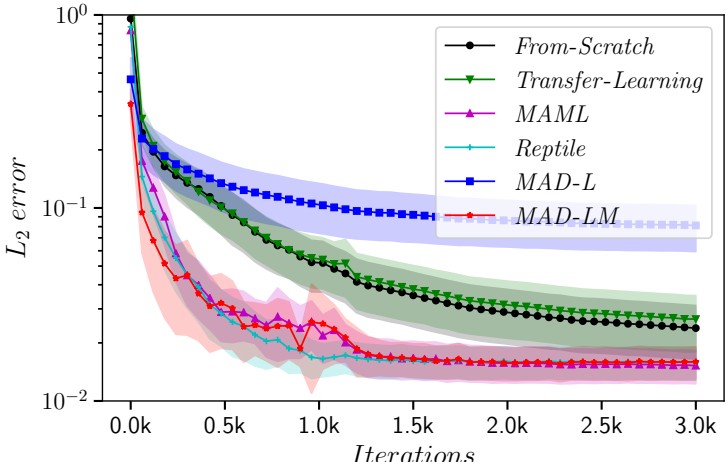

Figure 3: **Burgers' equation:** The mean $L_2\ error$ convergence with respect to the number of training iterations under heterogeneous PDE parameters.

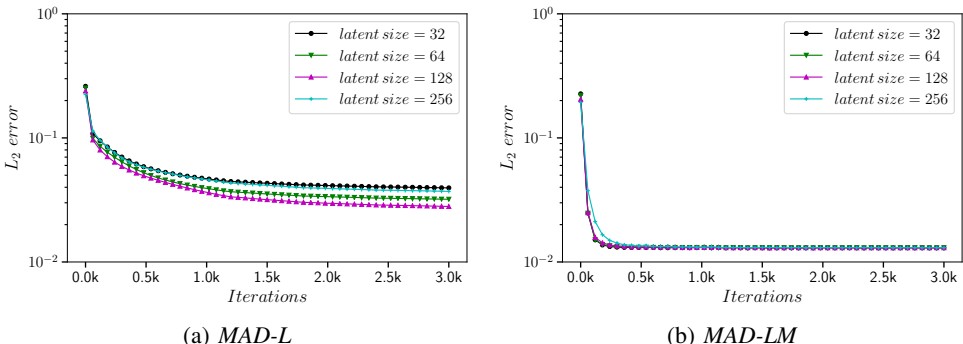

(a) *MAD-L*              (b) *MAD-LM*

Figure 4: **Burgers' equation:** The mean $L_2\ error$ of *MAD-L* (a) and *MAD-LM* (b) convergence with respect to the number of training iterations under different latent size.

Fig.9(b), for MAD-LM, although these latent sizes are quite different, they achieve very close performance.

## D   Detailed Experimental Setup and Extended Results for Time-Domain Maxwell's Equations

Except for the difference in equation coefficients $\epsilon_r$ and $\mu_r$, the settings of solution domain $\Omega$, initial conditions, boundary conditions and point source term $J$ are the same as those in [26]. Specifically, the solution domain $\Omega$ is $[0, 1]^2 \times [0, 4e-9]$. The initial electromagnetic field is zero everywhere and the boundary condition is the standard Mur's second-order absorbing boundary condition [30]. $J$ in Eq.14 represents a known source function and we set it to a Gaussian pulse. In temporal, this function can be expressed as:

$$J(x,y,t) = e^{-(\frac{t-d}{\tau})^2}\delta(x-x_0)\delta(y-y_0). \tag{4}$$

where $d$ is the temporal delay, $\tau$ is a pulse-width parameter, $\delta(\cdot)$ is the Dirac function used to represent the point source, and $(x_0, y_0) = (0.5, 0.5)$ is the location of the point source, $\tau = 3.65 \times \sqrt{2.3}/(\pi f)$ and the characteristic frequency $f$ is set to be $1GHz$. To solve the singularity problem caused by the point source, we use the $\delta(\cdot)$ function approximation method and the lower bound uncertainty weighting method proposed by [26]. In addition, the MS-SIREN network structure proposed by

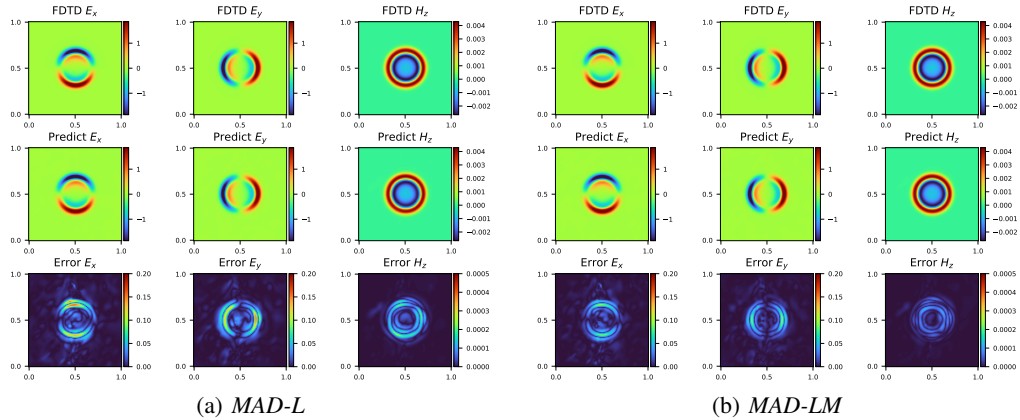

|     |     |
| :-: | :-: |
| (a) *MAD-L* | (b) *MAD-LM* |

Figure 5: **Maxwell's equations:** Model predictions of *MAD-L* (a) and *MAD-LM* (b) vs. the FDTD solutions at $t = 4ns$. **Top:** The numerical results of $(E_x, E_y, H_z)$ by FDTD method; **Middle:** The predicted $(E_x, E_y, H_z)$ through the deep learning model; **Bottom:** The absolute error between model predictions and the reference solutions.

[26] is used. To measure the accuracy of the model, the reference solution is obtained through the finite-difference time-domain (FDTD) [30] method.

We collect 25 pairs of $(\epsilon_r, \mu_r)$ at equal intervals from the region of $[1, 5]^2$ and randomly select 20 samples as $S_1$ with the rest 5 samples as $S_2$. For the training of *From-Scratch*, the pre-training and fine-tuning of *Transfer-Learning*, we set the total number of iterations to 100k. For the pre-training and fine-tuning of *MAD-L* and *MAD-LM*, we set the total iterations to 200k and 100k, respectively.

The instantaneous electromagnetic fields at time $4ns$ of *MAD-L* and *MAD-LM* compared with the reference FDTD results when $(\epsilon_r, \mu_r) = (3, 5)$ are depicted in Fig.10(a) and Fig.10(b), respectively. By observing the absolute error between the model predictions and the reference FDTD results, we can see that *MAD-LM* can achieve a lower absolute error. Specifically, the $L_2$ *error* of *MAD-L* is 0.037 and that of *MAD-LM* is 0.030.

We also apply *PI-DeepONet* to the solution of time-domain Maxwell's equations with a point source. In this experiment, the branch net of *PI-DeepONet* is a 4-layer fully connected network with 64 neurons in each hidden layer. The trunk net is an MS-SIREN [26] network, which consists of 4 subnets, each with 7 fully connected layers and 64 neurons in each hidden layer. We take the PDE parameter $(\epsilon_r, \mu_r)$ directly as the input of branch net. Because there are 3 output functions $(E_x(x, y, t), E_y(x, y, t), H_z(x, y, t))$, we adopt the method proposed in [31] to solve the multi-output problem, i.e. split the outputs of both the branch net and the trunk net into 3 groups, and then the $k$-th group outputs the $k$-th solution. However, due to the optimization difficulties caused by the singularity of the point source, *PI-DeepONet* appears to be very poor in accuracy (mean $L_2$ *error* is 0.672), while the mean $L_2$ *error* of *MAD-LM* is 0.028.

## E   Detailed Experimental Setup and Extended Results for Laplace's Equation

We generated 150 samples using GRF and randomly selected 100 samples as $S_1$, the remaining 50 cases are used as $S_2$. It should be declared that each sample corresponds to a different $h$ inside the boundary circle. For each sample, we randomly select $16 \times 1024$ points from the interior of the triangle and obtain the analytic solutions corresponding to these points to evaluate the accuracy of the model. When we apply the MAD method to solve this problem, it is not convenient to measure the distance between $\eta_{\text{new}}$ and $\eta_i$ since the variable PDE parameter $\eta$ is the shape and boundary condition of the solution domain. Therefore, the average of $|S_1|$ latent vectors obtained in the pre-training stage is used as the initialization of $z$ in the fine-tuning stage. For the training of *From-Scratch*, the pre-training and fine-tuning of *Transfer-Learning*, we set the total number of iterations to 10k. For the pre-training and fine-tuning of *MAD-L* and *MAD-LM*, we set the total iterations to 50k and 10k, respectively.

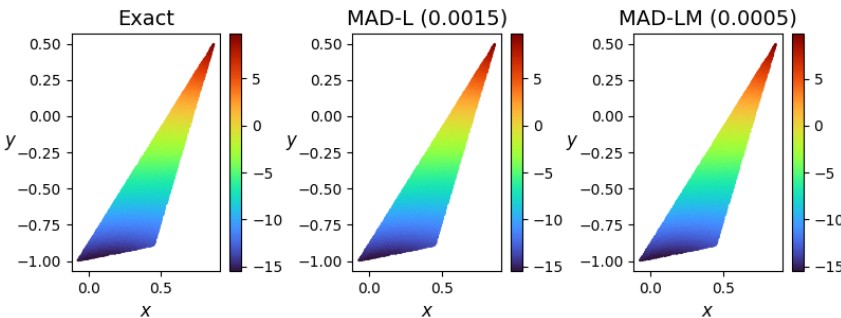

Figure 6: **Laplace's equation:** Analytical solutions, model predictions of *MAD-L* and *MAD-LM* (left to right).

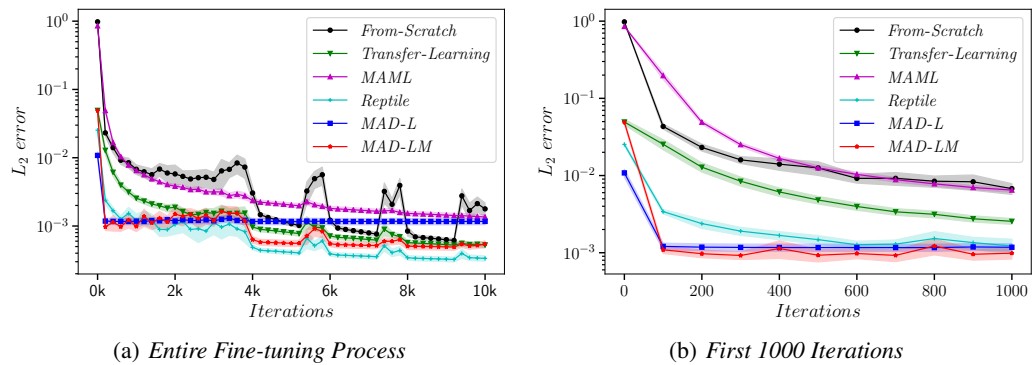

(a) *Entire Fine-tuning Process*

(b) *First 1000 Iterations*

Figure 7: **Laplace's equation:** The mean $L_2\ error$ convergence with respect to the number of training iterations when the solution domain $\Omega$ is a polygon with different shapes.

Fig.11 shows the predictions of *MAD-L* and *MAD-LM* compared with the analytical solutions under a randomly selected sample in $S_2$. To the naked eye, the prediction results of *MAD-L* and *MAD-LM* are almost identical to those of the analytical solution. However, the $L_2\ error$ of *MAD-L* is 0.0015, and that of *MAD-LM* is 0.0005.

We also consider a more complex scenario for Laplace's equation. Specifically, the shape of the solution domain $\Omega$ is a convex polygon arbitrarily taken from the interior of the unit circle. The number of sides of the polygon is in the range [3, 10] and the boundary conditions of the polygon

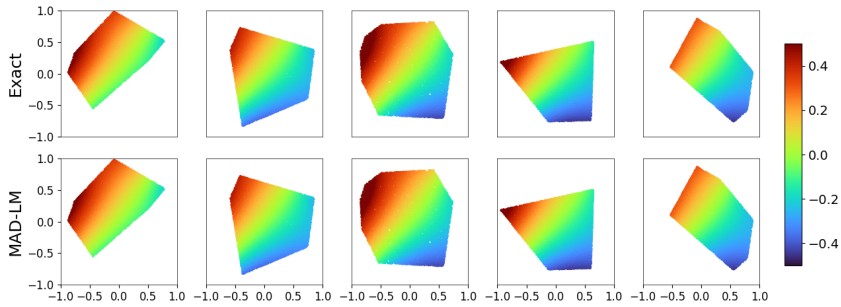

Figure 8: **Laplace's equation:** Analytical solutions, model predictions of *MAD-LM* when the solution domain $\Omega$ is a polygon with different shapes.

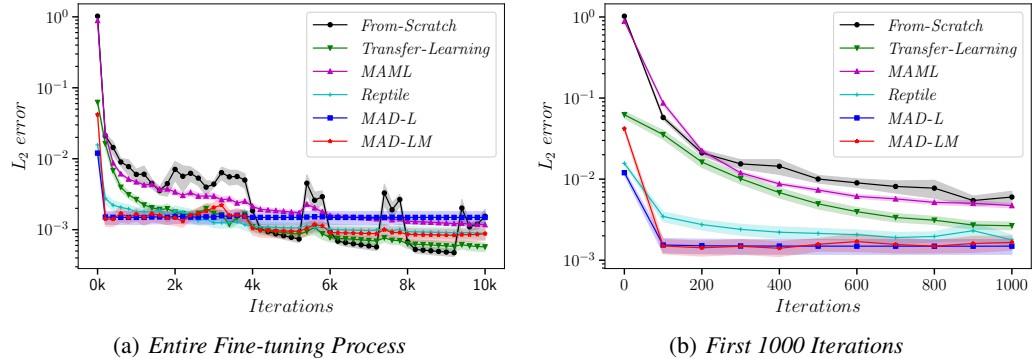

(a) *Entire Fine-tuning Process*  (b) *First 1000 Iterations*

Figure 9: **Laplace's equation:** The mean $L_2\ error$ convergence with respect to the number of training iterations for extrapolation experiments.

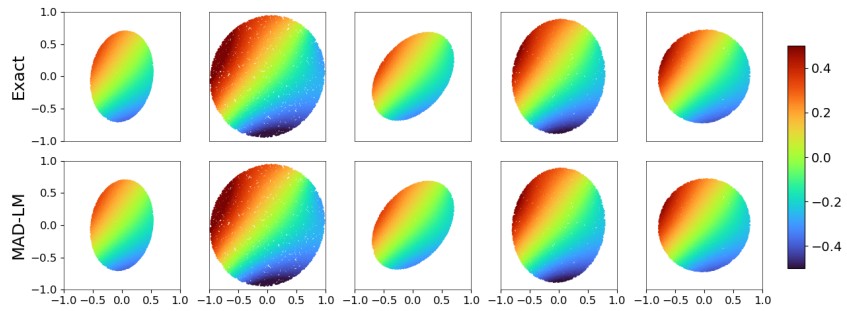

Figure 10: **Laplace's equation:** Analytical solutions, model predictions of *MAD-LM* for extrapolation experiments.

are generated in the same way as in Sec.3.3. It should be emphasized that the PDE parameters are heterogeneous for all experiments of Laplace's equation. Therefore, different solution domain shapes correspond to different $h$ and different $g(x, y)$. In this experiment, $|S_1| = 100$ and $|S_2| = 50$. Fig.12 compares the convergence curves of mean $L_2\ error$ corresponding to different methods, and Fig.12(a) shows the entire fine-tuning process and Fig.12(b) zooms in on the results of the first 1000 iterations. Compared to other methods, *MAD-L* and *MAD-LM* can achieve faster adaptation, i.e. very low $L_2\ error$ in less than 100 iterations. Fig.13 shows a comparison of the prediction of *MAD-LM* with the analytical solution under 5 randomly selected samples in $S_2$.

We also do an extrapolation experiment for Laplace's Equation. Specifically, in the pre-training stage, the shape of the solution domain $\Omega$ in $S_1$ is a convex polygon arbitrarily taken from the interior of the unit circle. However, in the fine-tuning stage, the shape of the solution domain $\Omega$ in $S_2$ is an ellipse arbitrarily taken from the interior of the unit circle. In this experiment, $|S_1| = 100$ and $|S_2| = 20$. Fig.14(b) shows that even in the case of extrapolation, *MAD-LM* can show faster adaptation compared to other methods. Fig.15 shows a comparison of the prediction of *MAD-LM* with the analytical solution under 5 randomly selected samples in $S_2$, which demonstrates the high accuracy of the solution obtained by *MAD-LM*.