# OpenReview forum: "Meta-Auto-Decoder for Solving Parametric Partial Differential Equations"
_NeurIPS.cc/2022/Conference — NeurIPS 2022 Accept_

### Official Review · Reviewer_1XCo · 2022-07-05

**Rating:** 6
**Confidence:** 3
**Soundness:** 3 good
**Presentation:** 4 excellent
**Contribution:** 2 fair

**Summary:**

This paper propose Meta-Auto-Decoder (MAD), a mesh-free and unsupervised deep learning method that uses pretraining by encoding PDE parameters to latent vectors. MAD has a manifold learning interpretation. Experiments show that MAD is faster than other deep learning-based methods.

**Questions:**

1. When pretraining, what data is used? In paper, it says 'randomly-generated'? Does it mean both target and pretrain data are randomly-generated? What knowledge is shared among these data?
2. When comparing effienciency, is the pretraining stage been taken into consideration for MAD?

**Ethics Review Area:**

["I don’t know"]

**Limitations:**

So far no.

**Strengths And Weaknesses:**

Strengths:
1. Well-motivated and clear presentation of contributions
2. Well-validated experimental result

Weaknesses:
1. Many concepts are shared with Meta-Mgnet and moreover not empirically compared with Meta-MgNet
2.  Pretraining part is a bit unclear to me (detailed in the questions part)

---

> ### Author Response · Authors · 2022-07-30
> **Answers to Questions Raised by Reviewer#3**
>
> ### Strengths And Weaknesses:
> **Q1**: Many concepts are shared with Meta-Mgnet and moreover not empirically compared with Meta-MgNet.
>
> **A1**: The only concept we share with Meta-MgNet is to view solving parametric PDEs as a meta-learning problem. In actual implementation, the difference is huge. Meta-MgNet is based on classical multigrid algorithm, and tested on elliptic PDEs of very specific types. In addition, unlike MAD, it requires to take the PDE parameter $\eta$ as network input, and hence needs to encode $\eta$ properly. For the specific types of the PDEs considered by Meta-MgNet, such encoding is relatively straightforward. But for the PDEs (e.g. the Laplace equation we considered), such encoding isn’t obvious. These make the empirical comparison with Meta-MgMet difficult.
>
> ---
>
> ### Questions:
> **Q2**: Pretraining part is a bit unclear to me (detailed in the questions part). When pretraining, what data is used? In paper, it says 'randomly-generated'? Does it mean both target and pretrain data are randomly-generated? What knowledge is shared among these data?
>
> **A2**:  The proposed MAD method is unsupervised. The “data” that we generate are in fact different PDE *tasks* given by different parameters $\eta$. The common knowledge that is shared by these tasks is that they all come from the same parametric PDE Eq.(1), or in other words, they associated to the solution manifold of the PDE with low-dimensional structures.
>
> ---
>
> **Q3**: When comparing efficiency, is the pretraining stage been taken into consideration for MAD?
>
> **A3**: The proposed MAD method mainly focuses on the applications in which $|S_2|\gg|S_1|$. Therefore, we take the fine-tuning speed to be the major consideration, and the cost of pre-training is not as important.

---

> > ### Comment · Reviewer_1XCo · 2022-08-09
> > **thanks for the informative reponse.**
> >
> > Hi, thanks for your reponse for resolving my concerns and I have raised my score. I suggest these clarifications will be incorporated into the final version to make it more clear.

---

> > > ### Author Response · Authors · 2022-08-09
> > > **Response to Reviewer's Suggestion**
> > >
> > > Thanks and we will update the paper accordingly.

---

### Official Review · Reviewer_Jrmb · 2022-07-10

**Rating:** 6
**Confidence:** 4
**Soundness:** 3 good
**Presentation:** 3 good
**Contribution:** 2 fair

**Summary:**

The paper proposes a method for learning the solutions of parameterized PDEs, where the PDE takes a set of parameters as an input. The paper introduces a neural network architecture (MAD) which takes coordinates as well as a hidden representation, which is implicitly encoded for each PDE parameters. After training, the proposed method solves optimization problem either to 1) find the best hidden representation or 2) find the best hidden representation and the best model parameters (i.e., weights and biases simultaneously).

**Questions:**

- how does the size of the latent vector affect the performance? how many samples (S1) would be required to learn a meaningful latent space?
- what is the computational overhead to solve the optimization problems in MAD-L and MAD-LM, respectively? (compared to other baselines)
- for the extrapolation task, how would the model performs if the testing parameters are generated in a following way: what if the training parameters are generated in a bounding box and the test parameters are chosen outside of the box.

**Limitations:**

the proposed method, in general, is technically sound, but would be more appreciated if they provide more in-depth analysis on the experimental results, e.g., providing intuitions for setting up the size of the latent dimension, the size of the decoder, the choice of optimizers, the choice of parameter set-ups, and so on. I believe all these choices would be PDE-specific and it would be great if the authors could provide some general guidelines for making choices on those matters. Also, it would be great to see whether this method can be generalizable to more complex PDE problems.

**Strengths And Weaknesses:**

[+] a simple, but effective idea to extend learning PDE solutions to parameterized PDE settings

[-] novelty: learning hidden representations of parameterized PDEs in training time and exploring the latent space for finding the best hidden representation for the new parameter instances have been studied in reduced-order modeling context (Eq (7.3) in Lee and Carlberg, JCP, 2020), which can be considered as an equivalent idea of MAD-L. Although the context and the optimization objective (Eq.(5)) are different, the idea of exploring latent space for the best hidden representation of parameterized PDEs has been out there.

---

> ### Author Response · Authors · 2022-07-30
> **Answers to Questions Raised by Reviewer#2 (Part III)**
>
> ### Limitations:
> **Q6**: the proposed method, in general, is technically sound, but would be more appreciated if they provide more in-depth analysis on the experimental results, e.g., providing intuitions for setting up the size of the latent dimension, the size of the decoder, the choice of optimizers, the choice of parameter set-ups, and so on. I believe all these choices would be PDE-specific and it would be great if the authors could provide some general guidelines for making choices on those matters.
>
> **A6**: Regarding the choice of the size of the latent dimension, we tried different sizes and took the best one. For Burgers' equation, please refer to A2. Regarding the choice of the size of the decoder, as we describe in Appendix B, our proposed method always performs better using different network architectures, and we chose a specific network architecture that can achieve high accuracy. Regarding the choice of optimizer, we also tried L-BFGS, but its performance was not as good as Adam for the three parametric PDEs solved in our paper. The choice of other hyperparameters was also determined based on trial and error.
>
> ---
>
> **Q7**: Also, it would be great to see whether this method can be generalizable to more complex PDE problems.
>
> **A7**: Please see A1 and A3 of reviewer#1.

---

> > ### Comment · Reviewer_Jrmb · 2022-08-08
> > **response to authors**
> >
> > I thank the authors for providing responses and the results of additional experiments.
> >
> > Re: Q1, the point mentioned in the question is not about latent variable modeling, which has been studied for more than 30 years in Computational Science, as indicated in the response. The comment is about a similarity in terms of ``solving an optimization problem over the (learned) latent space, searching for the best latent representation’’ in the test phase (again, solving the optimization problem in online phase as shown in Eq(7.3) in the referred paper).
> >
> > Re: Q2 and Q4, it would be informative if the authors could provide plots or tables showing performance (wall time/l2 error/etc) for varying latent dimensions.
> >
> > Assuming that these points to be properly addressed and added to the next version, I updated my rating.

---

> > > ### Author Response · Authors · 2022-08-09
> > > **Answers to Questions Raised by Reviewer#2 (Round II)**
> > >
> > > Thank you for your careful review of our work and constructive suggestions.
> > >
> > > **Q8**: For Q1, the point mentioned in the question is not about latent variable modeling, which has been studied for more than 30 years in Computational Science, as indicated in the response. The comment is about a similarity in terms of ``solving an optimization problem over the (learned) latent space, searching for the best latent representation’’ in the test phase (again, solving the optimization problem in online phase as shown in Eq(7.3) in the referred paper).
> > >
> > > **A8**: In the context of machine learning, optimizing the latent vector is another old idea, which goes back to at least (Tan and Mayrovouniotis, AIChE 1995), and is also utilized in the DeepSDF algorithm (our reference [22]). We apply such an idea to solve parametric PDEs, where the perspective of manifold learning (in infinite-dimensional spaces) indicates that fine-tuning $z$ only (MAD-L) could be insufficient. Therefore, we propose to fine-tune the model weights along with the latent vector, and take MAD-LM to be the more promising version of MAD. Experimental results confirm the superiority of MAD-LM over MAD-L, which only searches for the best latent representation as in previous work.
> > >
> > >
> > > **Q9**: For Q2 and Q4, it would be informative if the authors could provide plots or tables showing performance (wall time/l2 error/etc) for varying latent dimensions.
> > >
> > > **A9**: The URL ([https://files.catbox.moe/a13hl7.pdf](https://files.catbox.moe/a13hl7.pdf)) shows the mean $L_2\ error$ of MAD-L (a) and MAD-LM (b) convergence with respect to the number of fine-tuning iterations under different latent size for Burgers' equation. In fact, we have already answered this question in the previous A2. The table below shows the average wall-time required for one iteration under different latent sizes.
> > >
> > > | latent size | MAD-L's wall-time per iteration (second) | MAD-LM's wall-time per iteration (second) |
> > > | :---------: | :--------------------------------------: | :---------------------------------------: |
> > > |     32      |                 0.01952                  |                  0.02163                  |
> > > |     64      |                 0.02076                  |                  0.02168                  |
> > > |     128     |                 0.02092                  |                  0.02351                  |
> > > |     256     |                 0.02590                  |                  0.02485                  |

---

> ### Author Response · Authors · 2022-07-30
> **Answers to Questions Raised by Reviewer#2 (Part II)**
>
> **Q4**: What is the computational overhead to solve the optimization problems in MAD-L and MAD-LM, respectively? (compared to other baselines)
>
> **A4**: The following table shows the wall-time required for pre-training under different equations and different methods.
>
> |      methods      | Burgers' Eq (minute) | Maxwell's Eqs (minute) | Laplace's Eq (minute) |
> | :---------------: | :------------------: | :--------------------: | :-------------------: |
> | Transfer-Learning |         1.0          |         266.7          |          3.8          |
> |       MAML        |         11.0         |           -            |         63.3          |
> |      Reptile      |         45.2         |         6695.0         |         156.5         |
> |        MAD        |        1751.7        |        11830.0         |        1833.3         |
>
> Although the off-line pre-training of the MAD method takes more time compared to other methods, we set the goal to be similar to that of the foundation models in computer vision and natural language processing, and are mainly concerned with the speed of online fine-tuning on new tasks. The tables below show the number of iterations and wall-time required to fine-tune to a desired accuracy for different equations and different methods.
>
> * Burgers' Equation
>
>     The second column indicates the average wall-time required for one iteration, and the third column shows the number of iterations required to fine-tune to a desired mean $L_2 \ error$. The last column is the product of the previous two columns, as the average computational overhead corresponding to one sample (one PDE parameter). In this equation, MAD-L cannot reach the desired mean $L_2 \ error$.
>
>     |      methods      | wall-time per iteration (second) | iterations | total wall-time (second) |
>     | :---------------: | :------------------------------: | :--------: | :----------------------: |
>     |   From-Scratch    |             0.02039              |    660     |          13.457          |
>     | Transfer-Learning |             0.02018              |    750     |          15.135          |
>     |       MAML        |             0.01931              |    270     |          5.214           |
>     |      Reptile      |             0.02149              |    150     |          3.224           |
>     |       MAD-L       |             0.02092              |     -      |            -             |
>     |      MAD-LM       |             0.02351              |     90     |          2.116           |
>
> * Maxwell's Equations
>
>     |      methods      | wall-time per iteration (second) | iterations | total wall-time (second) |
>     | :---------------: | :------------------------------: | :--------: | :----------------------: |
>     |   From-Scratch    |             0.15935              |   61000    |         9720.35          |
>     | Transfer-Learning |             0.16310              |   43000    |         7013.30          |
>     |      Reptile      |             0.16394              |   61000    |         10000.34         |
>     |       MAD-L       |             0.18082              |    1000    |          180.82          |
>     |      MAD-LM       |             0.18713              |    8000    |         1497.04          |
>
> * Laplace's Equation
>
>     |      methods      | wall-time per iteration (second) | iterations | total wall-time (second) |
>     | :---------------: | :------------------------------: | :--------: | :----------------------: |
>     |   From-Scratch    |             0.02552              |    6200    |         158.224          |
>     | Transfer-Learning |             0.02599              |    6100    |         158.539          |
>     |       MAML        |             0.02544              |    8500    |         216.240          |
>     |      Reptile      |             0.02557              |    1700    |          43.469          |
>     |       MAD-L       |             0.02793              |    400     |          11.172          |
>     |      MAD-LM       |             0.02912              |    200     |          5.824           |
>
> ---
>
> **Q5**: For the extrapolation task, how would the model performs if the testing parameters are generated in a following way: what if the training parameters are generated in a bounding box and the test parameters are chosen outside of the box.
>
> **A5**: We have done such experiment in Section 3.2, where the pre-training parameters are sampled from the box $[1,5]^2$, and the fine-tuning parameter is $(7,7)\notin[1,5]^2$. In Section 3.1, the space of parameters $\mathcal{A}$ is an infinite-dimensional function space, in which the uniform distribution in an (infinite-dimensional) box cannot be defined. We thus take GRFs as the probability distribution, and use different GRFs to generate $S_1$ and $S_2$. Different distributions of domain shapes (convex polygons v.s. ellipses) are considered in Appendix E.

---

> ### Author Response · Authors · 2022-07-30
> **Answers to Questions Raised by Reviewer#2 (Part I)**
>
> ### Strengths And Weaknesses:
>
> **Q1** Novelty: learning hidden representations of parameterized PDEs in training time and exploring the latent space for finding the best hidden representation for the new parameter instances have been studied in reduced-order modeling context (Eq (7.3) in Lee and Carlberg, JCP, 2020), which can be considered as an equivalent idea of MAD-L. Although the context and the optimization objective (Eq.(5)) are different, the idea of exploring latent space for the best hidden representation of parameterized PDEs has been out there.
>
> **A1**: Finding a low dimensional representation of the solutions through offline pre-training and then computing the representation for a new instance through online adaptation is an old idea (20-30 years) in reduced-order modeling (e.g. the reduced basis method). However, the catch is how to execute this idea effectively in various applications. Classical reduced-order modeling uses linear representation which has limited performance. Thus, the community had been trying various nonlinear representations using DNNs for the past few years, where the work by Lee and Carlberg is a refined pioneering study.
>
> However, the proposed MAD method differs from Lee&Carlberg in many aspects. For example, MAD essentially learns low-dimensional representations in an infinite-dimensional function space, while Lee&Carlberg only considers finite-dimensional spaces. This is essentially due to the mesh-free representation used by MAD while Lee&Carlberg requires meshes to represent the solutions. Also, MAD does not require simulation data in its pre-training, while Lee&Carlberg does. MAD used a decoder-only architecture while Lee&Carlberg uses an encoder-decoder architecture.
>
> ---
>
> ### Questions:
> **Q2**: How does the size of the latent vector affect the performance?
>
> **A2**: We investigated the effect of the size of the latent vector (latent size) in Burgers' equation on performance. For the specific experimental results, please refer to: [https://files.catbox.moe/a13hl7.pdf](https://files.catbox.moe/a13hl7.pdf). As can be seen from Figure (a) in the link, for MAD-L, different latent sizes have different performances and the best performance is achieved when it is equal to 128. As can be seen from Figure (b), for MAD-LM, although these latent sizes are quite different, they achieve very close performance.
>
> ---
>
> **Q3**: How many samples (S1) would be required to learn a meaningful latent space?
>
> **A3**: For Burgers' equation, we discussed the effect of the number of samples $|S_1|$ in the pre-training stage on MAD-L and MAD-LM in Appendix C (Figure 7). Increasing the size of $S_1$ can improve the performance, but the benefit diminishes and may even reverse after the size reaches a certain point. This turning point depends on the specific type of the parametric PDE.

---

### Official Review · Reviewer_znr2 · 2022-07-11

**Rating:** 6
**Confidence:** 3
**Soundness:** 3 good
**Presentation:** 3 good
**Contribution:** 3 good

**Summary:**

This paper introduces a manifold learning approach for speeding up solutions to parametric PDEs, inspired by DeepSDF [22]. They associate a latent vector for each each instance of the PDE, and learn a solution manifold using an initial dataset of PDE instances. During inference, they either search for the closest latent vector (using gradient-based optimization) (MAD-L) or fine-tune both the network parameters alongside the latent vector (MAD-LM).

They report strong performance improvements compared to Meta-Learning, PINNs from scratch, and DeepONets.

**Questions:**

For the experiments with MAD, you had to introduce extra parameters in the NN to take the latent vector as input. Does this significantly increase parameter count? Could you give the number of NN parameters for "From-Scratch" experiments vs "MAD-L/LM" experiments?

It would be really nice to see a comparison of these methods to more standard numerical methods (such as FEM/spectral methods). it would be interesting to see if MAD-L can get close to or even surpass more classical methods in terms of performance.

Quality of life nits: Could you add descriptions of the figures into the captions themselves as well? It becomes difficult to scroll back and forth.

**Limitations:**

It was really nice seeing the cases where MAD-L failed. I think comparisons with more standard numerical methods could also make it clear where this method stands in terms of potential limitations compared to state of the art.

Potential negative societal impact was not assessed.

**Strengths And Weaknesses:**

The paper attacks an important problem. Often times we have to solve many versions of a PDE, either for design optimization or parameter discovery. One of the strengths of using neural networks as an ansatz for PDE solutions is that we can use techniques such as meta-learning to speed up solution process. This paper adapts methods from the shape representation literature to this problem, and shows consistent performance improvements.

The contribution is simple but works well, so I think it is solid. I like the sections on the manifold learning intuition, especially the simple ODE example, as I think it really motivates why the algorithm works.

It would be nice to see results from larger-scale experiments. Could you demonstrate a case-study where you would want to solve a sequence of PDEs, and your meta-learning setup gives large performance gains? I think this would help motivate the problem of parametric PDEs much better.

---

> ### Author Response · Authors · 2022-07-30
> **Answers to Questions Raised by Reviewer#1**
>
> ### Strengths And Weaknesses:
>
> **Q1**: It would be nice to see results from larger-scale experiments. Could you demonstrate a case-study where you would want to solve a sequence of PDEs, and your meta-learning setup gives large performance gains? I think this would help motivate the problem of parametric PDEs much better.
>
> **A1**: It is currently beyond the capability of our team. For large-scale experiments, we need a lot of computational resources to pre-train MAD and other meta-learning methods. The memory issue emerges as well since many coordinate points are sampled. Our paper aims to provide a new possibility, and present proof-of-concept experiments on some simple but demonstrative cases.
>
> ---
>
> ### Questions:
>
> **Q2**: For the experiments with MAD, you had to introduce extra parameters in the NN to take the latent vector as input. Does this significantly increase parameter count? Could you give the number of NN parameters for "From-Scratch" experiments vs "MAD-L/LM" experiments?
>
> **A2**: The increase of parameter count is relatively minor. The following is a comparison of the NN parameters of "From-Scratch" and "MAD-L/LM" for different problems:
>
> |   Problems    | From-Scratch | MAD-L/LM | Number (Ratio) of Extra Parameters |
> | :-----------: | :----------: | :------: | :--------------------------------: |
> |  Burgers' Eq  |    83073     |  99585   |           16512 (16.58%)           |
> | Maxwell's Eqs |    85004     |  89116   |            4112 (4.61%)            |
> | Laplace's Eq  |    83073     |  99585   |           16512 (16.58%)           |
>
>
> ---
>
> **Q3**: It would be really nice to see a comparison of these methods to more standard numerical methods (such as FEM/spectral methods). it would be interesting to see if MAD-L can get close to or even surpass more classical methods in terms of performance.
>
> **A3**: The three parametric PDEs tested in the paper are relatively simple, and MAD cannot outperform the standard numerical methods. However, it has the potential for more complex PDEs, especially when the dimension is high with complex boundaries so that generating high quality mesh and solving the underlaying algebraic equation is time consuming.  To conduct  experiments for such cases, however, requires computing resources that we cannot have.
>
> ---
>
> **Q4**: Quality of life nits: Could you add descriptions of the figures into the captions themselves as well? It becomes difficult to scroll back and forth.
>
> **A4**: Thank you for your suggestion. We will revise the paper accordingly in subsequent version of the paper.
>
> ---
>
> ### Limitations:
>
> **Q5**: It was really nice seeing the cases where MAD-L failed. I think comparisons with more standard numerical methods could also make it clear where this method stands in terms of potential limitations compared to state of the art.
>
> **A5**: Please refer to A3.

---

> > ### Comment · Reviewer_znr2 · 2022-08-07
> > **Response to response**
> >
> > Thanks for providing a response for the paper. After reading the other reviews, your responses, and the paper one more time, I do think the paper should be accepted, but would be much more improved with some sort of comparison to standard numerical methods or a larger scale experiment with a sequence of PDEs to really show the benefits of meta-training.
> >
> > Even if you do not think MAD cannot surpass classical numerical methods, it would still be interesting to understand how it stands in terms of performance. Ultimately the goal of PINN research is to surpass classical methods in some regime, and it is useful to know how small/large the gap currently is.
> >
> > Also, I wouldn't necessarily call a 16% increase in parameters "minor". It would be helpful to see comparisons with model sizes that are similar.

---

> > > ### Author Response · Authors · 2022-08-09
> > > **Answers to Questions Raised by Reviewer#1 (Round II)**
> > >
> > > Thank you for pointing out the core idea and contribution of this paper and approving our work.
> > >
> > > **Q6**: Even if you do not think MAD cannot surpass classical numerical methods, it would still be interesting to understand how it stands in terms of performance. Ultimately the goal of PINN research is to surpass classical methods in some regime, and it is useful to know how small/large the gap currently is.
> > >
> > > **A6**: The following table summarizes the wall-time to solve PDEs for a new PDE parameter:
> > >
> > > |      problems       | MAD-L (second) | MAD-LM (second) | Classical Numerical Method (second) |
> > > | :-----------------: | :------------: | :-------------: | :---------------------------------: |
> > > |  Burgers' Equation  |       -        |      2.116      |                0.481                |
> > > | Maxwell's Equations |     180.82     |     1497.04     |                0.912                |
> > > | Laplace's Equation  |     11.172     |      5.824      |                8.306                |
> > >
> > > For MAD-L and MAD-LM, all the three problems are solved on GPUs, and the iteration terminates after the desired $L_2 \ error$ is achieved. For the classical method, we use a split step method in Fourier spaces to solve Burgers' equation with MATLAB (Intel i5-9400F CPU, 2.9GHz). Laplace's equation is solved using the Fourier method with a basis size truncated to $N=100$, and Maxwell's equations are solved using FDTD method. The solvers of both Laplace's and Maxwell's equations are implemented using Python and Numpy, running on Intel Xeon E5-2966 CPU (2.3GHz). Our MAD method is relatively slow for Maxwell's equations, which is mainly due to the problem's singularity brought by the point-source. In Burgers' equation, MAD-L cannot reach the desired mean $L_2 \ error$.

---

### Meta-Review · Area_Chair_1X4e · 2022-08-27

**Recommendation:** Accept
**Confidence:** Less certain

**Metareview:**

The paper considers a meta-learning approach to solve families of PDEs (i.e. learning a solution operator). The main idea of the paper is to parametrize the solution of a member in the PDE family as being a function of a learned latent representation $z$ or that PDE. (Precisely, at a point $x$, they parametrize the solution as $f_{\theta}(x, z)$ where $\theta$ is globally learned, and $z$ is specific to a particular PDE.)
Beyond this, the parameters are fit as usual by minimizing some variational loss (e.g. $l_2$ error), and adapted to a new instance of a PDE as usual (by GD on an appropriately regularized $l_2$ loss).

The idea is fairly straightforward. It seems to perform well on small-scale data (e.g. families of Burgers' equation), compared to training from scratch and DeepONets, but comparisons on more challenging datasets are lacking. The authors also pointed out that fine-tuning (i.e. specializing to a new equation) is where the method shines --- which is reasonable given the parametric form that is fit.



**Award:**

No

---

### Decision · Program_Chairs · 2022-09-14

Accept